# *TERT* Promoter Alterations in Glioblastoma: A Systematic Review

**DOI:** 10.3390/cancers13051147

**Published:** 2021-03-08

**Authors:** Nathalie Olympios, Vianney Gilard, Florent Marguet, Florian Clatot, Frédéric Di Fiore, Maxime Fontanilles

**Affiliations:** 1Cancer Centre Henri Becquerel, Department of Medical Oncology, Rue d’Amiens, 76000 Rouen, France; Nathalie.olympios@chb.unicancer.fr (N.O.); florian.clatot@chb.unicancer.fr (F.C.); Frederic.difiore@chu-rouen.fr (F.D.F.); 2Neurosurgery Department, Rouen University Hospital, 1 rue de Germont, 76000 Rouen, France; Vianney.gilard@chu-rouen.fr; 3Department of Pathology, INSERM U1245, Normandie Univ, UNIROUEN and Rouen University Hospital, Normandy Centre for Genomic and Personalized Medicine, 76000 Rouen, France; florent.marguet@chu-rouen.fr; 4Inserm U1245, Normandie Univ, UNIROUEN, IRON Group, Normandy Centre for Genomic and Personalized Medicine, 76000 Rouen, France; 5Department of Digestive Oncology, Rouen University Hospital, 1 rue de Germont, 76000 Rouen, France

**Keywords:** glioblastoma, *TERT*, telomerase inhibition

## Abstract

**Simple Summary:**

Glioblastoma is the most common malignant primary brain tumor in adults. Glioblastoma accounts for 2 to 3 cases per 100,000 persons in North America and Europe. Glioblastoma classification is now based on histopathological and molecular features including *isocitrate dehydrogenase (IDH)* mutations. At the end of the 2000s, genome-wide sequencing of glioblastoma identified recurrent somatic genetic alterations involved in oncogenesis. Among them, the alterations in the promoter region of the *telomerase reverse transcriptase* (*TERTp*) gene are highly recurrent and occur in 70% to 80% of all glioblastomas, including glioblastoma *IDH* wild type and glioblastoma *IDH* mutated. This review focuses on recent advances related to physiopathological mechanisms, diagnosis, and clinical implications.

**Abstract:**

Glioblastoma, the most frequent and aggressive primary malignant tumor, often presents with alterations in the telomerase reverse transcriptase promoter. Telomerase is responsible for the maintenance of telomere length to avoid cell death. Telomere lengthening is required for cancer cell survival and has led to the investigation of telomerase activity as a potential mechanism that enables cancer growth. The aim of this systematic review is to provide an overview of the available data concerning *TERT* alterations and glioblastoma in terms of incidence, physiopathological understanding, and potential therapeutic implications.

## 1. Introduction

Glioblastoma is the most frequent and the most aggressive primary brain malignancy [1,2]. Glioblastoma is classified as grade IV glioma according to the WHO 2016 classification, which is the highest grade in the classification of gliomas. The standard of care for glioblastoma has not substantially changed in the past decade [3] and prognosis remains poor, with a median survival of approximately 15 months [4]. This is despite aggressive multimodal treatment combining surgical resection (when feasible), radiotherapy, and chemotherapy, which has a highly negative impact on the quality of life [5]. Glioblastoma was the first cancer to be studied by The Cancer Genome Atlas (TCGA), which aims to catalog and discover somatic genomic alterations in large cohorts of human tumors through integrated multidimensional analyses [6]. Molecular alterations mainly regarding *EGFR*, *ATRX*, *IDH1/2*, *TP53*, *ERBB2*, *NF1*, *PI3KR1*, and *MGMT* have been explored, paving the way for a new molecular era in neuro-oncology. Since these initial explorations, a variety of genetic and epigenetic alterations have been identified in glioblastoma [6,7,8] in an attempt to increase the molecular understanding of high-grade glioma pathogenesis and thus to personalize treatments to improve outcomes. Integrated genotypic and phenotypic data classifying central nervous system (CNS) tumors are now integrated in the 2016 updated WHO classification [9] to improve diagnosis. The 2016 classification distinguishes three categories of glioblastoma: non-mutated *IDH* glioblastomas, mutated *IDH* glioblastoma, and glioblastoma Not Otherwise Specified (NOS). Nevertheless, current findings have not yet led to subsequent changes in treatment modalities.

Among the potential drivers of interest, alterations in the *TElomerase Reverse Transcriptase* promoter (*TERTp*) have been reported in up to 80% of glioblastomas [10]. Telomeres are nucleoprotein complexes located at the end of chromosomes and are required for chromosomal integrity. Telomeres shorten at every cell cycle, eventually leading to cell death or senescence [11]. Telomerase is responsible for the repair of telomeres to maintain their length and avoid cell death. Telomere lengthening is required to achieve the infinite proliferation of cancer cells; thus, telomerase activity has been investigated as a potential mechanism for cancer growth [10]. Mutations in *TERTp*, the promoter region of *TERT* gene, were initially described in up to 70% melanoma tumors [12,13]. These mutations were then further explored in other tumors, including glioblastoma and additional emerging tumor research areas, such as hepatocellular carcinoma and urothelial carcinoma [10]. More precisely, mutations occur at two mutually exclusive hotspots located −124 bp and −146 bp upstream of the *TERT* translation start site: chromosome 5p15.33: 1,295,228 C > T and 1,295,250 C > T, referred to as C228T and C250T, respectively [14]. These mutations are well-described alterations and result in the upregulation of *TERT* expression required for telomerase activation [15].

The objective of this systematic review was to give an overview of available data concerning *TERTp* -124 and -146 alterations and glioblastoma in terms of incidence, physiopathological understanding, and potential therapeutic implications. Most of the present review will primarily focus on the two most common types of glioblastoma: “classical” non-mutated *IDH* glioblastoma as well as mutated *IDH* glioblastoma. A brief update will be also made on glioblastoma occurring in children and on rare histological subtypes of non-mutated *IDH* glioblastomas in adults.

## 2. Materials and Methods

A literature search was carried out to identify the relevant studies published since 2012. PubMed was searched for published articles, and clinicaltrials.gov (accessed on 20 December 2020) was searched for previous and ongoing clinical trials. Research terms included «glioblastoma», «glioma», «brain tumor» associated with «*TERT*», «telomeres», «telomerase», and «genomic landscape». The selected published studies included only original research papers; reviews were excluded. Additional relevant articles were identified from the reference list of articles identified in the initial search. This systematic review followed the Prisma statement.

## 3. Results and Discussion 

### 3.1. Flow-Chart

Overall, ninety-two studies were selected for the present review out of 586 pre-selected articles, Figure 1.

### 3.2. TERT Genomic Alterations in Glioblastoma

#### 3.2.1. Incidence

A comprehensive analysis of a TCGA data set found that among 6835 cancers, 73% expressed *TERT*. The *TERT*-expressing cancers were associated with *TERTp* mutations and with other point mutations, genomic rearrangements, DNA amplifications, or transcript fusions, and these alterations could predict telomerase activity [16]. Overall, *TERTp* mutations are the most frequent cancer genomic alterations. *TERTp* mutations occur in 51% of all glioma grades. Regarding glioblastoma, mutations commonly occur at two hotspots, referred to as C228T and C250T, which are mutually exclusive and occur in 80–90% of glioblastoma patients [10,17,18,19,20]. Such tumors most frequently have a frontal [21] or temporal location [22] and occur more frequently in older patients compared to *IDH*-mutated (*IDH*-mut) glioblastoma. Recently, two other *TERTp* gain-of-function alterations were described: *TERTp* c.1-100_1-79dup and *TERTp* c.1-110_1-89. These newly-described alterations occur in less than 1% of glioblastoma *IDH*-wild type (*IDH*-wt) and were not integrated into our systematic review [23].

#### 3.2.2. Diagnosis

The gold standard to identify *TERTp* mutations in glioblastoma remains based on molecular characterization of tumor DNA. The identification of *TERTp* mutations traditionally relied on Sanger sequencing, based on tumor DNA sequencing. Tumor heterogeneity or the scarcity of tumor DNA due to difficulties in tumor collection may lead to a lack of sensitivity of this historical technique [24,25,26,27]. Alternative sequencing methods were recently developed to increase the mutation detection rate in cases of low mutant allele frequency (MAF); these methods include Droplet Digital PCR (ddPCR), mass-spectrometry-based tests [28], and next-generation sequencing (NGS). ddPCR techniques have a higher sensitivity than Sanger sequencing in the detection of *IDH1* and *TERTp* mutations in glioma [29,30,31]. NGS assays also offer the possibility of deep detection and multiplexing the search for genomic alterations [32,33,34,35,36]. Euskirchen et al. [37] described a pocket-size nanopore sequencing device that could provide same-day detection of structural variants, point mutations, and methylation profiling. In contrast to the NGS assay, ddPCR-based *TERTp* mutation detection requires a lower tumor DNA quantity, and it might be useful in the peritumoral characterization of brain tumors [38]. Barritault et al. [39] applied molecular testing to 28 initially nondiagnostic biopsies of gliomas and were able to reclassify 6 of them after assessing for *IDH* and *TERTp* mutation status via SNaPshot PCR. The diagnostic performances of the different methods are summarized in Table 1.

The non- or minimally invasive detection of *TERTp* mutations is challenging in glioblastoma patients. It is desirable to increase diagnostic accuracy while limiting invasive procedures, especially for older patients and/or patients with a poor general condition. In this setting, the liquid biopsy concept has emerged in neuro-oncology. The concept of liquid biopsy is based on the molecular characterization of freely circulating tumor fragments that are found in easily accessible fluids such as plasma or cerebrospinal fluid (CSF).

Among these fragments, circulating tumor DNA is the most described in the literature and can be used to detect *TERTp* mutations by ddPCR or NGS-based sequencing methods [29,30,31,32,33,34,35,36,43]. ddPCR allows for significant sequencing depth and is thus suitable for the detection of a small quantity of circulating tumor DNA as observed in the plasma of patients suffering from glioma. The proportion of patients with a mutation detected in plasma is less than 10%; on the other hand, the specificity is 100% [31,44]. The low detection rate of *TERTp* mutations in plasma could be due to the size of the DNA fragments. These are shorter in patients with glioma and therefore may negatively influence the accuracy test of ddPCR by amplicon mismatch [45]. The plasma detection of *TERTp* mutations is an important issue for improving the management of patients with glioblastoma. Data on large prospective cohorts are still lacking, possibly linked to a limitation of current sequencing techniques.

The application of magnetic resonance imaging to diagnose glioblastoma and characterize the *TERTp* (wild type vs. mutated) status is also a noninvasive and promising technique [40,41,42,46,47,48]. In a recent study of 43 patients, Zhang et al. [42] showed that dynamic contrast-enhanced (DCE)-magnetic resonance imaging (MRI) histogram preoperative analyses demonstrate good analytical performance for the identification of *IDH, MGMT*, and *TERTp* alterations [42]. DCE-MRI provides qualitative and quantitative information on tumor perfusion. The mean ratio between the extravascular extracellular space and blood plasma, also called Kep, distinguished *TERTp*-mutated (*TERTp*-mut) glioma from *TERTp*-wild type (*TERTp*-wt) glioma with a sensitivity of 0.76 and a specificity of 0.78. Confirmatory and larger cohorts are now required to confirm the reproducibility of the results in order to generalize its use in daily practice. Invasive and non-invasive procedures to detect *TERTp* mutations are summarized in Figure 2.

### 3.3. Physiopathology

#### 3.3.1. Telomerase Activity: Overcoming Replicative Senescence

The limitless multiplication of cancer cells is a fundamental feature of cancer growth. Telomeres, which compose the terminal ends of each chromosome, are repetitive DNA sequences that protect chromosome ends from being recognized as double-strand breaks and therefore be destroyed by the DNA damage response system [49]. Telomeres shorten at every cycle, eventually leading to cell death.

To overcome this mechanism, cells can activate telomere-maintenance mechanisms such as telomerase activation [50]. Telomerase is a ribonucleoprotein that consists of an RNA subunit and a reverse transcriptase catalytic subunit, which adds telomeric repeat sequences of nucleic acids to chromosome ends, thereby maintaining telomere length [51]. Arita et al. [51] confirmed the somatic origin of *TERTp* mutations by sequencing 546 tumor samples and matched normal DNA from peripheral white blood cells in selected cases. Mutations in the *TERTp* region resulted in an ETS (E26 transformation-specific family transcription factor) binding site recognized by GABPA, a component of the multimeric transcription factor GABP, which facilitates reactivation of telomerase [52]. *TERTp* mutations activate *TERT* mRNA expression through the creation of a de novo ETS transcription factor-binding site [12,53]. The *TERT* expression level in tumors carrying *TERTp* mutations was found to be 6.1 times higher on average than that of wild-type tumors, indicating that the mutated promoter leads to *TERT* upregulation [51]. This finding was confirmed in further studies [7,53], suggesting that *TERT* expression represents a specific and sensitive surrogate for the presence of *TERTp* mutations.

*α-thalassemia/mental retardation syndrome X-linked (ATRX*) is an X-linked gene of the SWI/SNF family, mutations in which cause syndromal mental retardation and downregulation of a-globin expression [54]. ATRX and DAXX (death-associated protein 6) are central components of a chromatin-remodeling complex required for the incorporation of H3.3 histone proteins into the telomeric regions of chromosomes [55]. Dysfunction of the ATRX/DAXX complex is known to result in alternative lengthening of telomeres along with more widespread genomic destabilization. Interestingly, there is a significant inverse relationship between loss-of-function mutations in *ATRX* and *TERTp* in gliomas [51]. *ATRX*-mutated glioblastoma does not exhibit elevated *TERT* RNA expression compared to *TERTp*-mutated glioblastoma [7]. Alternatively, tumors without telomerase activity may acquire telomere lengthening by a homologous recombination-mediated mechanism known as alternative lengthening of telomeres (ALT). This suggests that *TERTp* mutations and alternative lengthening of telomeres secondary to *ATRX* mutations serve as complementary mechanisms for telomere lengthening and are an essential step in glioblastoma oncogenesis.

Whether *TERTp* mutations constitute an early or late event in glioblastoma genesis is yet not fully elucidated. Despite elevated *TERT* expression, *TERTp*-mutated tumors have shorter telomeres [16,53,56] than matched control samples, suggesting that these mutations may constitute a late event in oncogenesis when telomeres are exhausted. However, Abou et al. [57] suggested that glioblastoma develops early on from a common precursor with the loss of at least one gene copy (heterozygous deletion) of *PTEN* along with a *TERTp* mutation; this suggestion was based on their high frequency and their shared occurrence in different tumor foci from the same patient. A comparison of peritumoral tissue (subventricular zone), tumor tissue, and matched normal tissue found that the peritumoral area already harbored *TERTp* mutations and could be the origin of the tumor [58]. Korber et al. [59] suggested both a distant origin of de novo glioblastoma, up to seven years before diagnosis, and a common path of oncogenesis, with early occurrence of one or more chromosome rearrangements, such as 7 gain, 9p loss, or 10 loss. In this oncogenesis model, *TERTp* mutation occurs later, during the rapid growth of the glioblastoma.

Overall, the maintenance of telomere length via telomerase activity resulting from *TERTp* mutations appears to be an important event in gliomagenesis. The clinical impact of *TERTp* alterations, whether prognostic or therapeutic, is discussed later in this review.

#### 3.3.2. Association of *TERTp* Mutations and Other Molecular Alterations

The molecular characterization of glioblastoma has led to the identification of different prognostic glioblastoma subgroups based on the presence of *IDH* hotspot mutations, a well-established molecular feature of gliomas [60]; *TERTp* mutation and *MGMTp* methylation are also used for subgrouping. The classification of glioblastoma is now based on *IDH* status. The WHO 2016 classification distinguishes three entities: glioblastoma *IDH*-wt, glioblastoma *IDH*-mut, and glioblastoma Not Otherwise Specified—NOS. Glioblastoma NOS present with astrocytic features and anaplasia, microvascular proliferation, and/or necrosis but with unavailable *IDH* mutational status. *TERTp* mutations [9,17,61,62], *EGFR* alterations [24,63], and *MGMTp* methylation [14,21,64,65,66] are also integrated in daily practice. The majority of glioblastomas are *IDH-wt*; a combination of *TERTp* mutation and *IDH-wt* is the most common genotype observed in glioblastoma [14]. The differences in the biological processes involved in the telomerase pathway remain unclear between *IDH*-wt and *IDH*-mut glioblastomas. In comparison, a transcriptomic study carried out on samples of lower-grade gliomas did not make it possible to identify any differences between the *IDH*-mut/*TERTp*-wt and *IDH*-mut/*TERTp*-mut groups [67]. Beyond *IDH* and *TERTp*-mut glioblastoma, Diplas et al. [29] described a rare molecular subgroup of diffuse gliomas defined by the absence of common biomarkers (*IDH1/2*, codeletion 1p19q, *TERTp* mutations) and characterized by *SMARCAL1* inactivating mutations. *SMARCAL1* plays a role as a novel genetic mechanism of ALT and is involved in a novel mechanism of telomerase activation in glioblastomas that occurs via chromosomal rearrangement upstream of *TERT*. Integrating *TERTp* mutations into the landscape of molecular alterations in glioblastoma and clarifying the relationships among the known alterations could increase the molecular understanding of high-grade gliomas pathogenesis.

#### 3.3.3. *TERTp* Mutation Status: An Independent Prognostic Factor?

Whether *TERTp* mutation status is an independent prognostic factor is highly controversial. Numerous studies have highlighted the potential negative independent prognostic impact of *TERTp* mutations [18,19,20,21,24,25,61,62,68,69,70,71,72,73,74], whereas others [14,17,51,53,56,63,64,65,75,76] have suggested that the deleterious impact of *TERTp* mutation is correlated to the presence of cofounding molecular and clinical factors such as older age, *IDH-wt* status, and unmethylated *MGMTp* status. The *MGMT* gene codes for a protein involved in the DNA repair system. Its role is in particular to demethylate DNA, especially after alkylating agent exposition. Methylation of its promoter, and therefore gene silencing, occurs in approximately 45% of glioblastoma and is a favorable prognostic factor upon exposure to temozolomide. However, in adults suffering from glioblastoma, the presence of the methylation of the *MGMT* promoter does not currently constitute a biomarker necessary for the administration of temozolomide since temozolomide remains effective in overall survival in both methylated and unmethylated *MGMTp* glioblastomas. [66].

In a study of 473 adult gliomas among which 240 glioblastomas, Killela [62] studied both *IDH* and *TERTp* mutations and found that *TERTp* mutations in glioblastomas predicted poor survival even in tumors without an *IDH* mutation. Patients harboring *TERTp*-mut and *IDH*-wt tumors had the poorest overall survival (OS) of 11.3 months. Likewise, Labuissère [24] found that the presence of a *TERTp* mutation was an independent factor of poor prognosis (OS = 13.8 vs. 18.4 months), in both *IDH*-mut (OS = 13.8 vs. 37.6 months, *p* = 0.022) and *IDH*-wt glioblastomas (OS = 13.7 vs. 17.5 months, *p* = 0.006). Simon et al. [72] further suggested that not only was the presence of a *TERTp* mutation a significant negative predictor of OS but that *TERTp* mutations were prognostically relevant in patients with residual tumors who did not receive temozolomide chemotherapy, suggesting that surgery and temozolomide chemotherapy combined (in contrast to surgery plus radiotherapy) was effective against tumor cells responsible for the potentially adverse prognosis associated with *TERTp* mutations.

On the contrary, a large multivariable genomic analysis of 1122 gliomas among which 590 glioblastomas [53] failed to observe a statistically significant and independent survival association with the presence of a *TERTp* mutation after accounting for age and grade. Likewise, Pegmezi et al. [76] found that the presence of *TERTp* mutations was not independently associated with OS in an analysis of 1206 among which 360 were glioblastomas. Eckel Passow et al. [17] in a study of 1087 gliomas among which 472 glioblastomas found that in gliomas, *TERTp* mutations are generally unfavorable in the absence of *IDH* mutation and favorable in the presence of *IDH* mutation and 1p/19q codeletion.

Arita [14] analyzed the association between *TERTp*, *IDH* mutations, and *MGMTp* methylation status. *MGMTp* methylation is a well-established favorable prognostic factor for glioblastoma and is a predictive factor of response for elderly patients [66,77]. *TERTp* mutation was a favorable prognostic factor in *IDH*-mut glioblastoma, whereas it was an unfavorable prognostic factor in *IDH*-wt glioblastoma. *TERTp* mutation status appeared to depend not only on the IDH mutation status but also on the *MGMTp* methylation status in a combined cohort of 453 *IDH*-wt glioblastoma samples, where patients carrying *TERTp* mutations and unmethylated *MGMTp* had the poorest prognosis.

The overall survival results of the main studies are presented in Table 2. The prognostic role of *TERTp* mutations has not been clearly established since there are numerous confusing factors both clinical such as age, initial surgical procedure, and molecular such as *IDH* mutations, *MGMT* methylation status, or *EGFR* amplification. Prospective studies on large cohorts with a homogeneous patient population (for example glioblastoma *IDH*-wt and *MGMTp*-unmethymated) are still necessary to assess the independent prognostic impact of the *TERTp* mutation.

#### 3.3.4. Pediatric Glioblastoma

Brain tumors are the most common solid tumors in children and the leading cause of morbidity and mortality. Pediatric high-grade gliomas (pHGG) represent approximately 8 to 12% of pediatric brain tumors with a reported age-adjusted incidence of 0.26 per 100,000 population [78,79]. They mainly include diffuse astrocytic tumors, anaplastic astrocytoma, and glioblastoma [78,80]. They may manifest across all ages and anatomic CNS compartments [81]. Though phenotypically similar to adult glioblastoma, molecular profiling studies suggest a different biology in the pathogenesis of adult and pediatric high-grade gliomas [82,83].

Several pathways and molecular alterations were identified including the PI3K/AKT, Ras-Raf-MEK-ERK, RB, and p53 pathways as well as histone modifications [84]. In 2012, the first genome-wide sequencing among pediatric high-grade gliomas study identified a high frequency of alterations associated with histone modifications [85]. More specifically, mutations in the histone 3.1 and 3.3 proteins, encoded by the *HIST1H3B* and *H3F3A* genes, respectively. These alterations in the *H3.3-ATRX-DAXX* chromatin-remodeling pathway are present in 44% of glioblastomas and found to be specific to glioblastoma and highly prevalent in children and young adults. Subsequently, further studies have confirmed histone modifications as a hallmark of high-grade gliomas in children and young adults [81,83,86,87].

Mutations on H3.3 at G34 define a molecular subgroup of pHGG associating loss of function mutations in the tumor suppressor protein 53 (TP53) and mutations in ATRX or DAXX. ATRX and DAXX are components of a chromatin remodeling complex necessary for the incorporation of histone H3.3 at the pericentric heterochromatin of telomeres. ATRX inactivation is, therefore, necessary to result in telomerase-independent maintenance through ALT. This mechanism allows glioma cells to extend their telomeres without *TERT* expression and represents a way to avoid apoptosis thus enabling cancer progression [88]. BRAF (v-raf murine sarcoma viral oncogene homolog B1 gene/protein) Raf kinase-activating mutations occur in 5 to 10% of pHGGs, mutation *BRAF* V600E being the most commonly observed and associated with a significantly improved prognosis. *NTRK* fusion has also been described and reported in approximately 10% of non-brainstem pHGG and up to 40% of infants younger than three years. Mutations concerning either *BRAF* or *NTRK* fusions are of particular interest since they represent potentially targetable alterations. Data concerning inhibition has been encouraging in *BRAF* V600E mutant gliomas [89] as well as entrectinib in *NTRK*-fusion positive pHGG [90].

Data concerning *TERTp* mutations in pediatric glioblastoma are scarce. *TERTp* mutations were reported at a much lower rate in pediatric glioblastoma ranging from 3 to 11% [7,10,91] suggesting that infinite proliferation of cancer cells is generally not achieved by *TERTp* mutation-meditated activation of telomerase. Instead, they frequently display a loss of *ATRX* and an alternative lengthening of telomeres phenotype that maintains or increases telomere length [55,85]. Concerning the largest cohort of pHGG including midline pontine glioma, *TERTp* mutations were identified in 5/326 cases (1.5%). *TERTp* mutations were not associated with a histone mutation. Alternative lengthening of telomeres was mutually exclusive of *TERTp* mutations and present among 19.2% of only 26 analyzed samples [84].

Other mechanisms have also been reported such as methylation of the *TERTp* [92]. *TERTp* methylation located in a specific area—UTSS (upstream of the transcription start site)—was found to be a biomarker that can differentiate normal tissues and low-grade tumors from *TERT* expressing high-grade neoplasms. UTSS hypermethylation was associated with tumor progression and poor prognosis. Malignant tumors that did not have UTSS hypermethylation did not express *TERT* and had an *ALT* phenotype. In a study among 50 high-grade glioma samples [93], high *TERC* (telomerase RNA template) and *hTERT* expression were found in a majority of both brainstem and diffuse intrinsic pontine glioma. In multivariable analyses, increased *TERC* and *hTERT* levels were associated with worse prognosis in patients with non-brainstem high-grade gliomas, after controlling for tumor grade or resection extent. However, the prognostic relevance of *TERT* associated alterations in pediatric glioblastomas remains understudied.

*TERTp* alterations are very rare in pHGG compared to adult glioblastoma. Their association with histone alterations or IDH mutations as well as their clinical impact are still unresolved questions.

#### 3.3.5. Rare Tumors Subtypes

Glioblastoma *IDH*-wt does not constitute a homogeneous entity [94] and rare subtypes were described and included in a 2016 WHO classification [9]: gliosarcoma, giant cell glioblastoma, and epithelioid glioblastoma.

Gliosarcoma accounts for around 2% of glioblastomas and is composed of both a sarcomatous and glial component. Gliosarcoma patients are predominantly middle-aged men with a tumor frequently located in the temporal lobes. Its clinical particularity is its propensity to develop extracranial metastasis, which has been reported in up to 11% of patients [95]. Treatment of gliosarcoma encompasses the same approach as regular primary glioblastoma. It has been traditionally associated with poor prognosis but its prognostic significance still remains uncertain. As far as genetic alterations are concerned, common genetic alterations have been found in both sarcomatous and glial components suggesting a monoclonal origin [95,96]. Gliosarcoma is characterized by the absence of *IDH1/2* mutations, *TERTp* mutations in over 80% of cases, frequent *TP53* mutations, and absence of *ATRX* mutations and *EGFR* amplifications [97,98,99]. In a series of 36 gliosarcomas [97], *TERTp* mutations were reported in 88% of gliosarcoma and, when present, expressed in both the sarcomatous and glial components in 95% of cases. Overall, the mutations observed in gliosarcoma, apart from the absence of *EGFR* amplifications, are typical of “classical” *IDH*-wt glioblastoma (i.e., without the sarcomatous component).

Giant cell glioblastoma accounts for 1–5% of glioblastoma and is characterized by the presence of multinucleated giant cells with abundant eosinophilic cytoplasm. It occurs more frequently in younger adults around 45 years [97,100,101]. Giant cell glioblastoma survival is superior to that observed with glioblastoma especially with some patients experiencing longer survival. Indeed, a five-year survival rate of more than 10% has been described [100]. Consequently, it has been suggested that genetic differences may contribute to the improved survival of patients. Giant cell glioblastomas are characterized by the absence of *IDH1/2* mutations, high incidence of *TP53* mutations (80–90%), and *PTEN* mutations, frequent *ATRX* mutations, and rare *EGFR* amplifications. Data concerning *TERTp* mutations are scarce but mutations have been reported in 25–40% of cases [97,102]. Interestingly, microsatellite instability was reported in 30% of patients in two series of 12 [101] and 14 [102] giant cell glioblastomas suggesting that patients harboring those tumors may benefit from the use of immunotherapy.

Epithelioid glioblastoma is one of the most rare subtypes and was recently included in the last WHO Classification [9]. It occurs frequently in the first three decades of life and is distinguished histologically by epithelioid cells with abundant cytoplasm, prominent nucleoli, and rhabdoid cells [103]. Epithelioid glioblastomas have a very specific genetic background with an absence of *EGFR* amplification, *IDH1* gene mutations, or *PTEN* deletion, but instead, about half of them harbor *BRAF V600E* mutations [104,105]. *TERTp* mutations have not been described in the published series but a co-occurrence of *BRAF V600E* and *TERTp* mutation has been reported in a case report [106]. A recent study highlighted that epitheloid glioblastoma could be separated into three distinct subgroups based on their alterations and clinical profiles: *IDH*-wt glioblastoma-like tumors, anaplastic pleomorphic xanthoastrocytoma, and *RTK1* pediatric glioblastoma-like tumors [107].

Recently a new entity has emerged: diffuse astrocytic glioma, *IDH*-wt, with molecular features of glioblastoma, also called molecular glioblastoma. This entity appears with immunohistochemistry and/or iconographic features as a diffuse or anaplastic astrocytoma but the presence of specific molecular alterations (*IDH1/2* wildtype, *EGFR* amplification, whole chromosome 7 gain/whole chromosome 10 loss, and mutation of *TERTp*) reclassify the lesion as a grade IV glioma [38]. No difference in survival was demonstrated between patients carrying this *IDHwt* WHO grade IV astrocytoma and “classical” *IDH*wt glioblastoma: median OS 23.8 months vs. 19.2 months, *p* = 0.242 [108]. However, the optimal therapeutic strategy for these patients remains to be established in therapeutic trials dedicated to this population.

### 3.4. Therapeutic Implications and Perspectives

Considering the high frequency of *TERTp* mutation across glioblastoma and the fact that normal cells have lower telomerase activity than cancer cells, telomerase-inhibiting therapies appear to be an attractive target. However, due to the physiopathology of shortening telomeres, such a strategy is expected to be efficacious after multiple cell cycles in the presence of *TERT* inhibition. Currently, such targeted therapies are not approved in cancer care. Different approaches to target *TERT* activity, such as small molecule inhibitors, immunotherapy, and vaccines, are under investigation. Bajaj et al. [109] recently issued a review that encompasses biochemistry prerequisites for targeting telomerase, advantages, and challenges as well as actual and future development of telomerase inhibitors in solid tumors in general.

Regarding other usual chemotherapies, a preclinical study showed that eribulin, a microtubule inhibitor frequently prescribed in metastatic breast cancer, inhibited the growth of *TERTp*-mutated glioblastoma cell lines and significantly prolonged the survival of mice harboring brain tumors [110]. Eribulin has been described as a microtubule inhibitor but has also been shown to have specific inhibitory activity against a *TERT*—RNA-dependent RNA polymerase (RdRP) [111]. RdRP is described as one of the non–canonical functions of *TERT* and is involved in M-phase progression through the promotion of heterochromatin assembly and the maintenance of the stem-cell property.

Imetelstat is a small *TERT* inhibitor that has shown promising results in the treatment of essential thrombocytopenia, a chronic myeloproliferative neoplasm. Imetelstat was investigated for its ability to target early megakaryocyte progenitors and cancer stem cells because these cells have higher telomerase activity and shorter telomere lengths [112]. In glioblastoma cell lines [113], long-term imetelstat treatment led to progressive telomere shortening, reduced proliferation rates, and induced cell death in glioblastoma tumor-initiating cells. Imetelstat in combination with radiation and temozolomide had a dramatic effect on cell survival and activated the DNA damage response pathway. However, a clinical trial testing imetelstat in pediatric refractory CNS tumors was prematurely halted due to the death of two patients due to intratumoral hemorrhage secondary to treatment-related thrombocytopenia [114]. Further clinical trials are ongoing for both solid and hematological neoplasms in adults and younger patients. Pediatric tumors of interest include brain tumors, lymphomas, and refractory solid tumors. In adults, imetelstat is currently under investigation for myeloma, lymphoma, myelofibrosis as non-small-cell lung cancer, and breast cancer.

Mutations in the *TERTp* region result in an ETS site recognized by GABPA, a component of the multimeric transcription factor GABP, which facilitates reactivation of telomerase. The GABP transcription factor is an obligate multimer consisting of the DNA-binding GABPα subunit and trans-activating GABPβ subunit. GABPβ1L is a tetramer-forming β1L isoform of GABP that has been deemed necessary to activate the mutant *TERT* promoter in cells. Targeting GABPβ1L rather than *TERT* itself may represent a way to target *TERTp*-mutated cells while sparing normal cells to avoid the hematopoietic side effects observed with imetelstat. In glioblastoma cell lines, it was shown that disruption of the β1L isoform of GABP reverses the replicative immortality of *TERTp*-mutated glioblastoma cells [115]. In a mouse xenograft model of glioblastoma, knocking down GABPβ1L impaired tumor growth and increased mouse survival.

Preclinical data suggest that BIBR1532, a potent telomerase inhibitor, can induce apoptosis by downregulating telomerase activity at the transcriptional and translational levels [116,117]. However, to date, there are no available clinical data or ongoing clinical trials investigating BIBR1532.

Other strategies rely on the development of telomerase-targeted immunotherapy among which are *TERT* activity-targeted vaccines. Such an approach requires the identification of a tumor-associated antigen. An ideal tumor-associated antigen should have the following characteristics: a selective and broad expression in cancer cells, within all phases of tumor progression and the capacity to induce strong and effective immune responses. *hTERT* has been identified as such a tumor-associated antigen [118,119]. Dendritic cells (DC) represent a heterogeneous family of immune cells that link innate and adaptive immunity. They represent the most potent antigen-presenting cells in the human immune system and therefore constitute an effective tool to induce potent antitumor immune responses. In a phase I/II trial on seven glioblastoma patients [120], DCs transfected with RNA purified from autologous cancer stem cell cultures in combination with *hTERT* and mRNA were administered after the completion of standard post-operative chemo-radiotherapy. All treated subjects developed an immune response without significant toxicity or signs of autoimmunity. Vaccinated patients had significantly longer PFS compared to the historical-matched controls (694 days vs. 236 days, *p* = 0.0018) and 5/7 patients were alive after a two-year follow-up.

More recently, a phase II clinical trial [121] evaluated a cell vaccine (DCV) pulsed with glioblastoma stem-like cell antigens. Forty-three recurrent (*n* = 19) or primary (*n* = 24) glioblastoma patients were randomized at a 1:1 ratio after surgery to receive either DCV or placebo. Patients were stratified based on the mutational status of *IDH1/2* and *TERTp*. DCV did not significantly improve OS or PFS in all 43 patients. After adjusting for *TERTp* and *IDH1/2* mutational status, and B7-H4 expression, the DCV improved OS (*p* = 0.02; HR 2.5; 95% [CI] 1.15–5.45) but not PFS. B7 molecules are important mediators of immune evasion in the tumor microenvironment, among which B7-H4 is highly expressed in high-grade gliomas [122]. B7-H4 activation in the microenvironment of gliomas has been identified as an important immunosuppressive event blocking effective T-cell immune responses [123]. Interestingly, the *IDH1*-wt *TERTp-mut* glioblastoma subtype exhibited lower expression of B7-H4 compared with the other two groups, which could explain why *IDH1-wt TERTp-mut* glioblastoma patients showed a more important clinical benefit from DCV treatment. Glioblastoma patients presenting with *TERTp* mutated tumors may therefore constitute preferential candidates for DCV treatment.

UCPVax is a therapeutic anti-cancer vaccine based on telomerase-derived helper peptides designed to induce strong Th1 CD4 T cell responses [124]. This vaccine was reported to be safe in phase I trials though results have yet to be published. Phase II trials are ongoing in metastatic NSCLC both in monotherapy (NCT02818426) and in combination with nivolumab (NCT04263051). A phase I/II trial in adult patients suffering from glioblastoma is also ongoing (NCT04280848).

Considering the multiple pathways leading to telomerase activation, telomerase-inhibiting strategies offer the possibility to explore therapeutic strategies as diverse as vaccines, immunotherapies, and the reconsideration of standard chemotherapies. Physiopathological consequences of *TERTp* mutations and potential druggable targets are shown in Figure 3.

To date, there are no validated and efficient glioblastoma treatments regarding *TERT*p mutations.

## 4. Conclusions

Telomere maintenance mechanisms during DNA replication are essential across glioblastomas. *TERTp* mutations are the most represented alterations in glioblastoma, suggesting a pivotal role in oncogenesis. The identification of *TERTp* mutations is essential and is currently integrated into glioblastoma diagnostic procedures. Despite data from multiple sources, the prognostic impact of *TERTp* mutations remains controversial. A better understanding of the molecular mechanisms underlying *TERTp*-mutated glioblastoma could lead to the development of *TERT*-targeted therapies. Preclinical and clinical trials are ongoing, but no such therapy has yet demonstrated clinical efficiency in glioblastoma patient care.

## Figures and Tables

**Figure 1 cancers-13-01147-f001:**
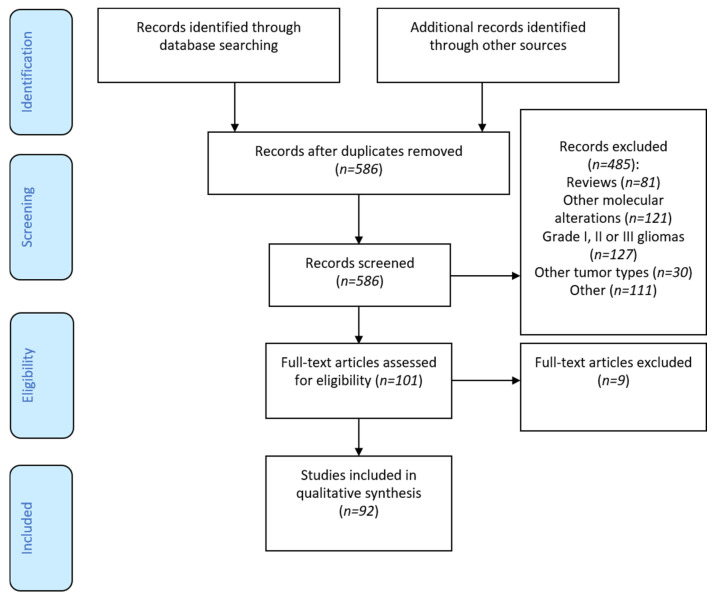
Flow-chart.

**Figure 2 cancers-13-01147-f002:**
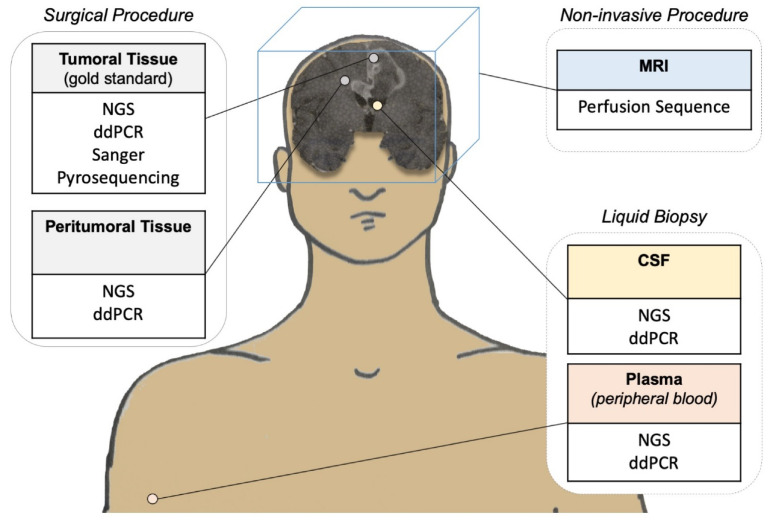
Methods to detect *TERTp* alterations in patients with glioblastoma. The current reference method is the detection of alterations based on sequencing techniques applied to tumor tissue obtained from either lesion resection or from targeted biopsy in cases of non-resectable tumors. The other methods, particularly noninvasive or minimally invasive methods, are still under development and are not yet used in routine clinical practice.

**Figure 3 cancers-13-01147-f003:**
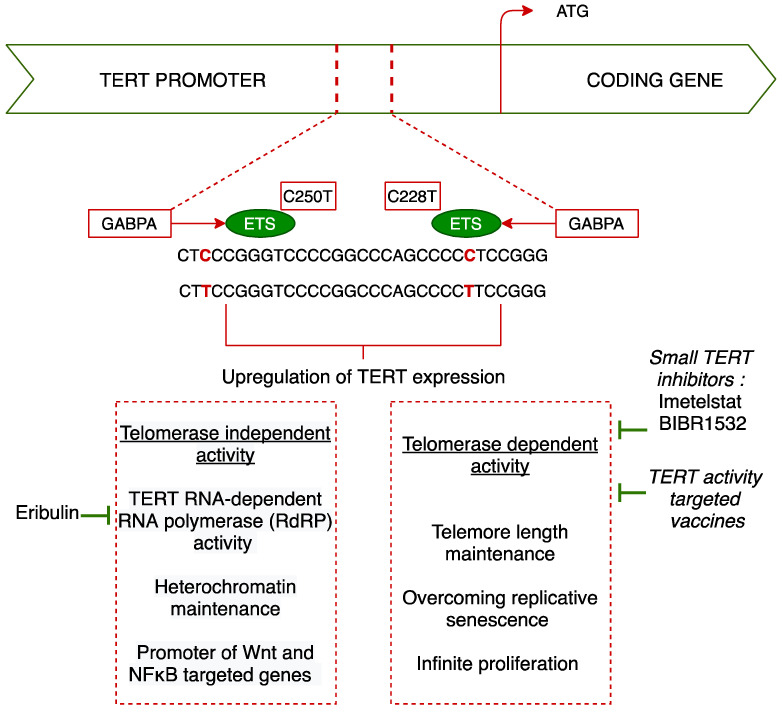
Physiopathological consequences of *TERTp* alterations in glioblastoma and therapeutic perspectives.

**Table 1 cancers-13-01147-t001:** Diagnostic performances of the different methods to detect *TERTp* mutations.

Method	Population (Number of Patients)	Reference Method	Accuracy	Reference
Molecular biology techniques on the tumor				
NGS targeted panel	18 glioblastomas	-	7 *TERTp* mut/18 (38.9%)	[32]
NGS targeted panel	47 glioblastomas	Sanger sequencing	30 *TERTp* mut/47 (64%) Se 99%, Spe 100%	[34]
NGS targeted panel	121 gliomas	Sanger sequencing	66 *TERTp* mut/121 Se 100%, Spe 100%	[36]
Nanopore	16 glioblastomas	NGS	Se 100%, Spe 60%	[37]
Droplet digital PCR	52 grade IV gliomas	Sanger sequencing	Se 100%, Spe 100%	[31]
Molecular biology techniques on the periphery of the tumor				
Snapshot	22 gliomas	NGS	Se 87.5%, Spe 100%	[39]
MRI parameters				
Support Vector Machine	112 gliomas	Tumor sequencing	Se 85.7%, Spe 54.8%	[40]
Spectroscopy	112 gliomas	Tumor sequencing	Se 83.3%, Spe 95.2%	[41]
Dynamic susceptibility contrast- and dynamic contrast-enhanced- MRI	60 gliomas	Tumor sequencing	Se 56–84%, Spe 53.6%–83.3%	[42]

**Table 2 cancers-13-01147-t002:** Prognostic impact of *TERTp* mutations in glioblastoma.

Population (Number of Patients)	*TERTp-mut* vs. *TERTp-wt* Glioblastoma (Median Overall Survival, Months)	Independent Factor?	Reference
453 *IDH*-wt glioblastomas	14.6 vs. 18.8	**Uncertain** Confounding factor with *MGMTp* methylation	[14]
303 *IDH*-wt glioblastomas	18.5 vs. 17.8, *p* = 0.3845	**No**	[64]
358 glioblastomas (322 [89.9%] *IDH*-wt)	9.6 vs. 9.3, *p* = 0.22	**No** Association with *IDH* mutation	[75]
395 *IDH*-wt glioblastomas	13.7 vs. 17.5, *p* = 0.006	**Uncertain** Confounding factor with *EGFR* amplification	[24]
178 *IDH*-wt glioblastomas	11 vs. 16, *p* = 0.038	**Uncertain** Association with tumor resection and exposition to temozolomide	[72]
243 *IDH*-unknown glioblastomas	10 vs. 21, *p* < 0.001	**Uncertain** No stratification on *IDH* statusAssociation with *TERT* polymorphism rs2853669	[73]

## Data Availability

Not applicable.

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
