# Peer review of "TERT Promoter Alterations in Glioblastoma: A Systematic Review"

_cancers, 2021, doi:10.3390/cancers13051147_

Round 1

Reviewer 1 Report

Authors wrote a systematic review fully reporting biological and clinical features related to TERT promoter mutations in glioblastoma. The review is well structurated and well written. The authors discussed the topic exhaustively. They underlied the putative clinical impact of TERT promoter mutation in the risk stratification of patients, in predicting sensitivity towards specific compounds, and also as markers to design specific treatment.

Minor revisions:

INTRODUCTION

Lines 49-52. This statement should me modified in order to cite also two rare pathogenetic mutations of TERT promoter that have been recently described in glioblastoma (Pierini T, et al. Acta Neuropathol Commun. 2020). The relevance of these tandem duplication at TERTp, although rare, relies on their occurrence in other human cancer, i.e. oligodendroglioma, myelodisplastic syndromes, papillary thyroid carcinoma. ….“Although mutations mainly occur at two mutually ………, two new pathogenetic variants consisting of tandem nucleotide duplications, have been recently reported….”

3.3.3

Line 206-207. Simon et al found that TERTp mutations were prognostically relevant (instead of particularly) in………

3.3.5

Lines 224-225. It is not clear what the authors mean by… astrocytic tumors… ..beginning the sentence talking about anaplastic astrocytomas and glioblastomas, which are astrocytic tumors; Please clarify this point. They also wrote "both malignant" a statement which makes no sense as astrocytomas are malignant tumors

3.3.6.

Lines 304-307. This discussion is not the purpose of the review. In my opinion the sentence should be removed

Author Response

Authors wrote a systematic review fully reporting biological and clinical features related to TERT promoter mutations in glioblastoma. The review is well structurated and well written. The authors discussed the topic exhaustively. They underlied the putative clinical impact of TERT promoter mutation in the risk stratification of patients, in predicting sensitivity towards specific compounds, and also as markers to design specific treatment

We warmly thank the reviewer for this comment.

Minor revisions:

INTRODUCTION

Lines 49-52. This statement should be modified in order to cite also two rare pathogenetic mutations of TERT promoter that have been recently described in glioblastoma (Pierini T, et al. Acta Neuropathol Commun. 2020). The relevance of these tandem duplication at TERTp, although rare, relies on their occurrence in other human cancer, i.e. oligodendroglioma, myelodisplastic syndromes, papillary thyroid carcinoma. ….“Although mutations mainly occur at two mutually ………, two new pathogenetic variants consisting of tandem nucleotide duplications, have been recently reported….”

The reviewer is right and the referred article was recently published. Our review focused on the two recurrent TERTp-124 and -146 alterations. However, two sentences were modified in the Introduction section to clarify this point (lines 55 and 58, page 2) and a sentence referring to the new alterations and the above-mentioned article was added in the 3.2.1 section lines 135-138, page 3.

3.3.3

Line 206-207. Simon et al found that TERTp mutations were prognostically relevant (instead of particularly) in………

The sentence is now corrected in the 3.3.3 section line 294 page 7.

3.3.5

Lines 224-225. It is not clear what the authors mean by… astrocytic tumors… ..beginning the sentence talking about anaplastic astrocytomas and glioblastomas, which are astrocytic tumors; Please clarify this point. They also wrote "both malignant" a statement which makes no sense as astrocytomas are malignant tumors

We thank the reviewer for this comment: the sentence is now modified to anaplastic astrocytoma and glioblastoma; and diffuse astrocytic tumors, in the 3.3.4 line 328 page 8.

3.3.6.

Lines 304-307. This discussion is not the purpose of the review. In my opinion the sentence should be removed

The reviewer is right; the conclusion of 3.3.5 section on rare grade IV glioma is now modified as well as the Conclusion of this section.

Reviewer 2 Report

This review article addresses TERT promoter mutations and their significance in glioblastoma. The authors address a number of topics related to TERT, including its function in telomere maintenance, clinical testing modalities to detect TERT mutations, prevalence in adult and pediatric glioblastoma, prognostic significance, and therapies targeting TERT. The authors perform an in depth literature review. However, given the breadth of this analysis, much of the discussion is quite superficial. The article could be improved by providing a more focused and in-depth analysis, perhaps focusing primarily on IDH-wildtype glioblastoma in adults. Additionally, the authors seem to have a somewhat incomplete understanding of brain tumor classification and glioblastoma subtypes.

Major points:

The aspects of the review that discuss brain tumor genetics and classification are confusing. In particular:

  1. The authors do not provide a clear discussion of adult glioblastoma classification with respect to IDH-mutant vs. IDH-wildtype tumors and the differing mechanism these entities use to maintain their telomeres. Similarly, there is not clear discussion of MGMT for readers which may be unfamiliar with this prognostic marker.
  2. The discussion of giant cell glioblastoma, gliosarcoma, and epithelioid glioblastoma is not up to date and only peripherally related to the topic of TERT. In the 2016 WHO classification, giant cell and gliosarcoma are considered subtypes of IDH-wildtype glioblastoma. Recent studies suggest that epithelioid glioblastoma is not a single entity (PMID: 28990704).
  3. The discussion of pediatric gliomas is also confusing. The authors should address this topic with respect to different molecular groups of pediatric high-grade glioma (e.g. diffuse midline glioma, H3 G34 mutant tumors, BRAF mutant, RTK driven tumors). Which pediatric glioma types utilize TERT for telomere maintenance? Is there any research to suggest why pediatric high-grade gliomas more frequently harbor ATRX mutation?
  1. The discussion of the prognostic significance of TERT should be expanded and synthesized. To simply say that the data are controversial is not helpful to readers. Two specific papers are discussed in detail. Why were these papers chosen for this discussion? What is the authors’ opinion regarding the prognostic significance of TERT? The discussion of TERT as a predicative factor includes a single paper with a result that may be a false positive--why would TERT C250T vs. C228T be different in their significance?

The discussion of therapeutic implications raises several questions:

  1. a. Does the mechanism of eribulin specifically involve TERT? Similarly, does the pulsed DCV specifically target TERT?
  2. Are Imetelstat and BIBR1532 the only TERT inhibitors in development? If a recent review is available on this topic, it would be helpful to point this out.
  3. What is GABPb1L and how does this relate to TERT? This is not explained in the text?

The detection of TERT by NGS or other techniques is routinely used in clinical practice. The discussion of other emerging imaging and cell free methods is more interesting and could be expanded.

  1. How sensitive and specific are methods for detection of TERT in cell-free DNA?
  2. With respect to MRI techniques, what features differentiate TERT mutant vs. TERT WT glioblastomas? Why was only one paper discussed in any detail in this section?

Author Response

This review article addresses TERT promoter mutations and their significance in glioblastoma. The authors address a number of topics related to TERT, including its function in telomere maintenance, clinical testing modalities to detect TERT mutations, prevalence in adult and pediatric glioblastoma, prognostic significance, and therapies targeting TERT. The authors perform an in depth literature review. However, given the breadth of this analysis, much of the discussion is quite superficial. The article could be improved by providing a more focused and in-depth analysis, perhaps focusing primarily on IDH-wildtype glioblastoma in adults. Additionally, the authors seem to have a somewhat incomplete understanding of brain tumor classification and glioblastoma subtypes.

We thank the reviewer for this comment and we fully agree with the reviewer. A concise description of the WHO 2016 classification is now integrated in the 3.3.2. section lines 256-260 page 7. This point now clarifies the fact that the most common and described glioblastoma are glioblastoma IDH-wt. In our review the mention ‘glioblastoma’ is more used because the use of the 2016 WHO classification is not often used in the in the referred studies. If we have wished to specifically focus on glioblastoma IDH-wt and selected only studies referring to the 2016 WHO classification many key-articles would have been not selected. On the other hand, if we had stated that the default subtype was 'glioblastoma IDH-wt' instead of ‘glioblastoma’ in all articles, we would probably have made mistakes in interpreting selected articles.

Major points:

The aspects of the review that discuss brain tumor genetics and classification are confusing. In particular:

  1. The authors do not provide a clear discussion of adult glioblastoma classification with respect to IDH-mutant vs. IDH-wildtype tumors and the differing mechanism these entities use to maintain their telomeres.

The reviewer is right; differences involved in biological processes to maintain the telomerase activity are not yet fully elucidated between TERTp mut/IDH-wt glioblastoma and TERTp-mut/IDH-mut glioblastoma. This point is now mentioned in the 3.3.2 section lines 264-267 page 7.

Similarly, there is not clear discussion of MGMT for readers which may be unfamiliar with this prognostic marker.

The referee is right and more detailed description of MGMT and the prognostic impact of the methylation of its promoter is now added in the 3.3.3 section lines 282-285, page 7.

  1. The discussion of giant cell glioblastoma, gliosarcoma, and epithelioid glioblastoma is not up to date and only peripherally related to the topic of TERT. In the 2016 WHO classification, giant cell and gliosarcoma are considered subtypes of IDH-wildtype glioblastoma. Recent studies suggest that epithelioid glioblastoma is not a single entity (PMID: 28990704).

The reviewer is right: sentences were added along with the above-mentioned reference and a conclusion sentence were added in the 3.3.5 section lines 421-427 page 10.

  1. The discussion of pediatric gliomas is also confusing. The authors should address this topic with respect to different molecular groups of pediatric high-grade glioma (e.g. diffuse midline glioma, H3 G34 mutant tumors, BRAF mutant, RTK driven tumors). Which pediatric glioma types utilize TERT for telomere maintenance? Is there any research to suggest why pediatric high-grade gliomas more frequently harbor ATRX mutation?

We thank the reviewer for this remark. Precise data on TERTp mutations in pediatric high-grade gliomas depending on different molecular groups does not exist to our knowledge. Interactions between histone modifications and ATRX mutations are now explicated in the 3.3.4 section lines 342-356 page 8-9.

  1. The discussion of the prognostic significance of TERT should be expanded and synthesized. To simply say that the data are controversial is not helpful to readers. Two specific papers are discussed in detail. Why were these papers chosen for this discussion? What is the authors’ opinion regarding the prognostic significance of TERT? The discussion of TERT as a predicative factor includes a single paper with a result that may be a false positive--why would TERT C250T vs. C228T be different in their significance?

We thank the reviewer for this comment and we agree. The discussion on the prognostic significance of TERTp mutations was enriched with further details on the most important publications (in the 3.3.3 section lines 286-307 page 10). The publications chosen for description correspond to the largest series and/or highly valued publications. As far as the predictive significance is concerned, data seemed indeed too scarce to further develop. The paragraph on predictive impact is now fully removed.

The discussion of therapeutic implications raises several questions:

  1. a. Does the mechanism of eribulin specifically involve TERT? Similarly, does the pulsed DCV specifically target TERT?

We thank the reviewer for this comment. Concerning eribulin, the mechanism has been explicated in the 3.4 section lines 443-447 page 10.

Pulsed DCV does not specifically target TERT though TERTp mutated tumors appear to be more sensitive to DCV. This point is now added in the 3.4 section lines 507-508 page 11.

  1. Are Imetelstat and BIBR1532 the only TERT inhibitors in development? If a recent review is available on this topic, it would be helpful to point this out.

Imetelstat and BIBR1532 are to our knowledge the TERT inhibitors in development in Neuro-Oncology. A review on current development of TERT inhibitors in general has been added in the section 3.4 lines 436-439 page 10.

  1. What is GABPb1L and how does this relate to TERT? This is not explained in the text?

Further explanations were added section 3.4 lines 463 to 467 page 11.

The detection of TERT by NGS or other techniques is routinely used in clinical practice. The discussion of other emerging imaging and cell free methods is more interesting and could be expanded.

  1. How sensitive and specific are methods for detection of TERT in cell-free DNA?

The referee is right and liquid biopsy is a challenging topic in neuro-oncology. A more detailed description of the technique is now added in the 3.2.2 section lines 170-196 page 5.

  1. With respect to MRI techniques, what features differentiate TERT mutant vs. TERT WT glioblastomas? Why was only one paper discussed in any detail in this section?

A sentence is now added in the 3.2.2 section lines 185 – 190 page 5. Only one paper is discussed as regard to the size of its cohort and its methodological quality. A sentence to discuss the need of confirmatory and larger cohorts is now added.

Round 2

Reviewer 2 Report

In this revision, the authors have added additional discussion, but this does not substantially address two of my main concerns with the article. 

  • My biggest concern is that the authors do not seem to have a clear understanding of current brain tumor classification. There is considerable discussion on this topic and the authors do not come across as being experts in this area. In order for this paper to be published, much more effort needs to be put into addressing this. Perhaps the authors could involve an additional co-author such as a neuropathologist who would have additional expertise in this area. Below are some examples.

-The text added in lines 256-259 is not accurate.  There is abundant literature showing that IDH-mutant astrocytomas use ATRX to maintain telomeres. IDH-mutant low-grade gliomas are clearly defined by histology and molecular genetics as astorcytomas (IDH-mutant, ATRX mutant) and oligodendrogliomas (IDH-mutant, TERT mutant, 1p/19q co-deleted).

-The additional discussion of MGMT does not address its main significance—predicting response to temozolomide.

-The discussion on gliosarcoma and giant cell glioblastoma remains unchanged from the previous version.  This discussion has very little to do with TERT and is not accurate. For example: “it appears that giant cell glioblastomas share characteristics with both primary and secondary glioblastomas occupying and intermediate position.” What exactly does this mean?  Primary and secondary glioblastoma terminology is no longer used. Another example: “the molecular alterations of TERTp and their role for the other histological subtypes are quite similar with IDH-wt glioblastoma.”  Giant cell and gliosarcoma are subtypes of IDH-wildtype glioblastoma.

-The discussion of pediatric glioblastoma also has little to do with TERT.  The authors imply that there is little information regarding TERT in pediatric HGG, which is simply not true.  There have been many large-scale studies on the genetics of pediatric HGG, which include assessment of TERT mutation status.  For example, this study analyzed 1,000 pediatric HGG (PMID: 28966033).

  1. The discussion of the prognostic significance of TERT has been expanded, but remains difficult to follow. Perhaps the authors could include a table or some other way to summarize this information. Currently, it is impossible to draw any meaningful conclusions from this discussion. I appreciate that this is a complex issue, but the current discussion does not adequately address the topic.

Author Response

In this revision, the authors have added additional discussion, but this does not substantially address two of my main concerns with the article. 

  • My biggest concern is that the authors do not seem to have a clear understanding of current brain tumor classification. There is considerable discussion on this topic and the authors do not come across as being experts in this area. In order for this paper to be published, much more effort needs to be put into addressing this. Perhaps the authors could involve an additional co-author such as a neuropathologist who would have additional expertise in this area. Below are some examples.

We thank the reviewer for this comment on the WHO classification 2016 which is an important point. Throughout this new version of the review we have tried to clarify this point. However, two critical difficulties persist: (i) it is not always referred to the WHO 2016 classification in some cited papers and it does not seem appropriate to us to delete them because of their use of the old classification (WHO 2007); (ii) it is difficult to clearly distinguish between the common type IDH wild type glioblastoma (ex glioblastoma multiforme) from the grade IV glioma subtypes which also include rare histological subtypes (gliosarcoma, giant cell glioblastoma). We choose to call the common ‘glioblastoma IDH wild type’ ‘classical’ in this new version. An effort was made in this new version (for example Lines 35-36, 50-52 and lines 72-76) to clarify our thinking. As recommended by the reviewer, the manuscript was also proofread and corrected by an experienced neuropathologist from the Rouen university hospital (Florent Marguet, MD, PhD) as well as two experienced oncologists (Florian Clatot, MD, PhD; Prof. Frédéric Di Fiore, MD, PhD) now integrated in co-authors. As suggested by them, we have also introduced the designation “diffuse astrocytic glioma, IDH-wildtype, with molecular features of glioblastoma, WHO grade IV” also named “molecular glioblastoma” (lines 433-443 in the 3.3.5 section) which appeared in the WHO classification 2016 and which represents a real diagnostic challenge for the neuropathologist.

- The text added in lines 256-259 is not accurate.  There is abundant literature showing that IDH-mutant astrocytomas use ATRX to maintain telomeres. IDH-mutant low-grade gliomas are clearly defined by histology and molecular genetics as astorcytomas (IDH-mutant, ATRX mutant) and oligodendrogliomas (IDH-mutant, TERT mutant, 1p/19q co-deleted).

We thank the reviewer for this comment and we do agree. The sentence is now changed to ‘The clinical impact of TERTp alterations, whether prognostic or therapeutic, is discussed later in this review’ lines 248-249.

- The additional discussion of MGMT does not address its main significance—predicting response to temozolomide.

We thank the reviewer for this comment. A sentence referring to the prognostic impact of the methylation of the MGMT promoter already exists in the first version of the manuscript lines 312-313. Additional information on the prognostic impact are now added lines 284-289. However, we do not agree with the reviewer: promoter methylation of the MGMT gene is a favorable prognostic factor but not a predictive biomarker of temozolomide response. This point is now also discussed lines 284-289.

-The discussion on gliosarcoma and giant cell glioblastoma remains unchanged from the previous version.  This discussion has very little to do with TERT and is not accurate. For example: “it appears that giant cell glioblastomas share characteristics with both primary and secondary glioblastomas occupying and intermediate position.” What exactly does this mean?  Primary and secondary glioblastoma terminology is no longer used. Another example: “the molecular alterations of TERTp and their role for the other histological subtypes are quite similar with IDH-wt glioblastoma.”  Giant cell and gliosarcoma are subtypes of IDH-wildtype glioblastoma.

We do agree with the reviewer and the above-mentioned sentence is now removed. On the other hand, and when taking into account of the WHO 2016 classification, there is no term to clearly distinguish the type ‘glioblastoma IDH-wild type’ (which includes rare histological subtypes) from the common glioblastoma IDH-wild type or old ‘glioblastoma mutliforme’ in the previous classification. That’s why the term ‘classical’ is now used lines 407-408. We do hope that this point is now more clarified.

-The discussion of pediatric glioblastoma also has little to do with TERT.  The authors imply that there is little information regarding TERT in pediatric HGG, which is simply not true.  There have been many large-scale studies on the genetics of pediatric HGG, which include assessment of TERT mutation status.  For example, this study analyzed 1,000 pediatric HGG (PMID: 28966033).

We thank the reviewer for this comment and we agree that pediatric brain tumors are grandly different from adult high-grade gliomas. We still want to maintain our paragraph on this point to briefly recall that the carcinogenesis pathways encountered in pHGGs mainly differ from the pathways encountered in adults’ glioblastoma. Regarding TERTp alterations we find it interesting to mention that its role is much less important than in adults’ glioblastoma. The above-mentioned article is already in our bibliography (reference 84, Mackay et al). We have substantiated our discussion concerning the results of this important study lines 370-374 and we have modified the conclusion of this paragraph lines 387-389.

  1. The discussion of the prognostic significance of TERT has been expanded, but remains difficult to follow. Perhaps the authors could include a table or some other way to summarize this information. Currently, it is impossible to draw any meaningful conclusions from this discussion. I appreciate that this is a complex issue, but the current discussion does not adequately address the topic.

We thank the reviewer for this comment. TERTp mutations as an independent prognostic factor is still questioning since many studies identify confounding factors (association with MGMTp methylation or IDH status for examples). A table that summarizes main results of TERTp mutations on survival from large cohort studies is now added in the manuscript line 3éà. The end of the 3.3.3 is also modified to ‘The overall survival results of the main studies are presented in Table 2. The prognostic role of TERTp mutations has not been clearly established since there are numerous confusing factors both clinical such as age, initial surgical procedure and molecular such as IDHmutations, MGMT methylation status or EGFR amplification. Prospective studies on large cohorts with a homogeneous patient population (for example glioblastoma IDH-wt and MGMTp-unmethymated) are still necessary to assess the independent prognostic impact of the TERTp mutation.’ lines 323- 329.